

# Temperature and land use change are associated with *Rana temporaria* reproductive success and phenology

Kat E. Oliver and Xavier A. Harrison

Centre for Ecology and Conservation, University of Exeter, Falmouth, United Kingdom

## ABSTRACT

Chemical pollution, land cover change, and climate change have all been established as important drivers of amphibian reproductive success and phenology. However, little is known about the relative impacts of these anthropogenic stressors, nor how they may interact to alter amphibian population dynamics. Addressing this gap in our knowledge is important, as it allows us to identify and prioritise the most needed conservation actions. Here, we use long-term datasets to investigate landscape-scale drivers of variation in the reproductive success and phenology of UK Common frog (*Rana temporaria*) populations. Consistent with predictions, we found that increasing mean temperatures resulted in earlier initialisation of spawning, and earlier hatching, but these relationships were not consistent across all sites. Lower temperatures were also linked to increased spawn mortality. However, temperature increases were also strongly correlated with increases in urban area, arable area, and nitrate levels in the vicinity of spawning grounds. As with spawning and hatching, there was marked spatial variation in spawn mortality trends, where some sites exhibited steady increases over time in the proportion of dead or diseased spawn. These findings support previous work linking warming temperatures to shifts in timing of amphibian breeding, but also highlight the importance of assessing the effect of land use change and pollution on wild amphibian populations. These results have implications for our understanding of the response of wild amphibian populations to climate change, and the management of human-dominated landscapes for declining wildlife populations.

## INTRODUCTION

Anthropogenic stressors are altering ecosystems across the world (*Ceballos et al., 2015*), with chemical pollution, land cover change, and climate change being highlighted as significant drivers of declines in ecosystem functioning (*Walther et al., 2002*; *Thuiller et al., 2011*; *Howard, Flather & Stephens, 2020*). These altered ecosystems arise due to the effects of anthropogenic stressors on individual organisms, which can change community structures through decreased reproductive success and altered phenology (*Bison et al., 2021*). Although anthropogenic stressors influence a myriad of organisms, amphibians are particularly susceptible due to their unique life cycle and physiology (*Blaustein & Kiesecker, 2002*).

Corresponding author
Xavier A. Harrison,
x.harrison@exeter.ac.uk

Amphibia are subject to multiple threats including pathogens, land use change and climate change (*Hof et al., 2011*), and the combined effects of these processes have led to amphibians becoming the most threatened vertebrate group (*Hoffmann et al., 2010*). Amphibian breeding phenologies are advancing at a fast rate compared to other vertebrate taxa (*Parmesan, 2006*; *Cohen, Lajeunesse & Rohr, 2018*), suggesting they are particulalry sensitive to climatic change. However, the relative importance of environmental change, land use change and pollution as drivers of variation in breeding success, and how these factors may vary across different populations, remains unresolved. It is vital that we understand the drivers of changes in amphibian breeding timing and success, given the key role played by amphibians in many ecosystems (*Burton & Likens, 1975*; *Davic & Welsh, 2004*; *Mallory & Richardson, 2005*; *Wood & Richardson, 2010*).

Climate change has affected amphibian reproductive success and phenology in both positive and negative ways. Increased winter precipitation resulting from climate change can benefit some amphibians (*Benard, 2015*), but extreme weather events and advanced phenology are likely to negatively impact others (*Blaustein et al., 2010*; *Thuiller et al., 2011*; *Buss, Swierk & Hua, 2021*). In a review of the impacts of increased droughts and extreme precipitation, *Walls, Barichivich & Brown (2013)* conclude that even amphibian species adapted to variable environmental conditions are not able to adapt fast enough to keep up with dramatic changes in precipitation patterns. Drought negatively affects the reproductive success of amphibians (*e.g.*, *Piha et al., 2007*), whilst warming temperatures are responsible for trends towards earlier amphibian breeding (*Ficetola & Maiorano, 2016*). For example, warmer winter temperatures caused earlier breeding in *Rana sylvatica*, but also lower female fecundity (*Benard, 2015*), and have been associated with decreases in female body condition in female *Bufo bufo* (*Reading, 2007*). Recent work on the common spadefoot toad (*Pelobates fuscus*) found a negative correlation between initiation of breeding migration and temperature (*i.e.*, a delay in breeding with warmer temperatures; *Dalpasso et al., 2023*). It was suggested that spadefoot toads are likely using precipitation as a cue rather than temperature, highlighting the importance of assessing multiple axes of climatic variation when attempting to understand the timing of amphibian breeding (*Dalpasso et al., 2023*). Previous studies on *Rana temporaria* have linked changes in spring temperatures and climate with advanced breeding phenology in both the UK (*Carroll et al., 2009*; *Phillimore et al., 2010*) and mainland Europe (*Montori & Amat, 2023*). However, *Phillimore et al. (2010)* demonstrated marked heterogeneity among populations of *R. temporaria* in the magnitude of phenological shifts in spawning date over time, suggesting that between-population variation in spawning dates may be driven by local adaptation. Understanding how different populations respond to variation in climatic cues is vital for prioritising conservation efforts, and for generating accurate predictions of future responses to climate change at the landscape level.

Chemical pollutants can increase mortality and alter rates of development in amphibian species (*Carey & Bryant, 1995*), and the release of many chemical pollutants into the environment is growing (*Sharma et al., 2020*). The application of nitrate and ammonium fertilisers in the United States has increased by 4,000% since the 1940s

(*Cao, Lu & Yu, 2018*), which has led to environmental nitrogen levels being high enough to impact amphibians (*Rouse, Bishop & Struger, 1999*). Nitrogen fertilisers are toxic to amphibian embryos and larvae above certain levels, causing methemoglobinemia (*Huey & Beitinger, 1980*), and can also lead to eutrophication (*Boyer & Grue, 1995*). These effects can scale up to change reproductive success and phenology in amphibian populations. Ammonium nitrate exposure has been linked to species-specific survival in three Australian amphibian species (*Hamer et al., 2004*). Exposure of *Litoria aurea* larvae to 10–15 mg/l ammonium nitrate resulted in significantly reduced survival, but no such effect was seen in *Crinia signifera* or *Limnodynastes peronii* (*Hamer et al., 2004*). This suggests that the declines solely seen in *L. aurea* populations were due to the effect of ammonium nitrate on this species alone. Ammonium nitrate may also be responsible for declines in European species; *Hyla arborea*, *Discoglossus galganoi*, and *Bufo bufo*, all of which have reduced survival in <200 mg/l of ammonium nitrate (*Ortiz et al., 2004*). Ammonium nitrate lowers larval development rate of *Pleurodeles waltl*, *Bufo calamita*, and *Pelobates cultripes*, suggesting that sub-lethal doses of ammonium and nitrate ions could alter amphibian phenology by delaying metamorphosis (*Ortiz et al., 2004*). Recent work has shown that both ammonium and nitrate can slow the laerval development of *Alytes obstetricans*, and that high concentrations of ammonium can cause larval mortality (*Garriga, Montori & Llorente, 2017*).

The most likely driver of increased exposure to chemicals like ammonium nitrate is land cover change. Many amphibian species use terrestrial habitats to disperse from natal ponds (*Semlitsch, 2008*), meaning high-quality terrestrial habitat is needed to prevent isolation (*Marsh & Trenham, 2001*). Isolation can lead to reduced reproductive success in amphibian populations (*Allentoft & O'Brien, 2010*), so both habitat quality and connectivity are vital, but land cover change can alter both these factors. Previous studies have shown that urbanisation reduces the area of suitable habitat available to many amphibians (*Price, Browne & Dorcas, 2012*), as well as significantly increasing fragmentation and isolation (*Natuhara & Zheng, 2022*). Expansion of intensive agriculture can also lead to amphibian declines, with areas of Mediterranean cropland having significantly lowered amphibian abundance (*Beja & Alcazar, 2003*) compared to the surrounding countryside. The mechanisms driving these changes are likely the combined effects of reduced connectivity of habitat fragments, alongside increased exposure to chemicals. In a study on wood frogs (*Rana sylvatica*), *Buss, Swierk & Hua (2021)* found that climate-induced shifts in phenology (earlier *versus* later breeding) altered susceptibility to anthropogenic contaminants like NaCl, highlighting the importance of studying both breeding phenology and pollution in tandem. Despite this abundance of research on the impacts of individual anthropogenic stressors on amphibian reproductive success and phenology, few studies have evaluated the relative impacts of these stressors simultaneously. This deficit can lead to difficulties in identifying the most effective actions needed to conserve amphibian species and the ecosystems they belong to *Frick, Kingston & Flanders (2020)*. Here we investigate the relative impacts of chemical pollutants, land cover change, and climate change on the reproductive success and phenology of one amphibian species: *Rana temporaria*.
*R. temporaria* is an Anuran species widely distributed across Europe (*Sillero et al., 2014*) that breeds in still, fresh water (*Haapanen, 1982*) but spend the majority of time in terrestrial habitats (*Dabagyan & Sleptsova, 1991*). Although widespread, many *R. temporaria* populations are declining (*Cooke, 1972*; *Loman & Andersson, 2007*; *Guarino, Di Già & Sindaco, 2008*) and their breeding phenology is advancing (*Beebee, 1995*; *Sparks et al., 2007*; *Scott, Pithart & Adamson, 2008*; *While & Uller, 2014*; *Bison et al., 2021*; *Montori & Amat, 2023*). Like other amphibian species, a combination of chemical pollutants, land cover change and climate change are altering *R. temporaria* reproductive success and phenology. Although there are conflicting results as to whether ecologically relevant levels of ammonium and nitrate are directly lethal to *R. temporaria* spawn and larvae (*Oldham et al., 1997*; *Johansson, Räsänen & Merilä, 2001*), multiple studies demonstrate that high levels of these ions can reduce larval *R. temporaria* fitness, as well as delay metamorphosis (*Johansson, Räsänen & Merilä, 2001*; *Manson, 2002*; *Oromí, Sanuy & Vilches, 2009*). In contrast, land cover change has been shown to cause *R. temporaria* population declines due to reduced reproductive success. For example, *Cooke (1972)* attributes the national declines in UK *R. temporaria* populations between 1940 and 1970 to the draining of wetlands to make way for intensive agriculture and urbanisation. Similar impacts of agricultural expansion have been seen in Sweden, where cropland populations are declining (*Loman & Andersson, 2007*). These declines are not due to increased chemical pollutants (*Loman & Lardner, 2006*) but could be due to lack of the suitable terrestrial refugia that adults need (*Marnell, 1998*), leading the reduced gene pools and reproductive success (*Allentoft & O'Brien, 2010*). There is little evidence to suggest that climate change is causing similar declines, with the only example being a period of high temperatures leading to reduced female fecundity in France (*Neveu, 2009*). On the other hand, warming temperatures have been widely associated with earlier breeding phenology in *R. temporaria* (*Neveu, 2009*), with *Scott, Pithart & Adamson (2008)* finding that dates of breeding congregations, spawning, and hatching advanced in correlation with warming temperatures.

Here, we use a 21-year long-term monitoring dataset of *R. temporaria* breeding populations in the UK to quantify the relative importance of chemical pollutants, land cover change, and climate change on *R. temporaria* reproductive success and phenology. We test the following predictions:

1) Increasing average winter temperatures will result in earlier timing of both spawning and hatching, but the strength (slope) of these relationships will be population-specific

2) Land cover change, measured as the increases in arable and urban area, will be associated with decreased reproductive success.

3) Increased levels of pollutants and lower temperatures during spawning will be associated with greater degrees of spawn mortality (measured as the percentage of dead or disease eggs (*Rennie et al., 2017*).

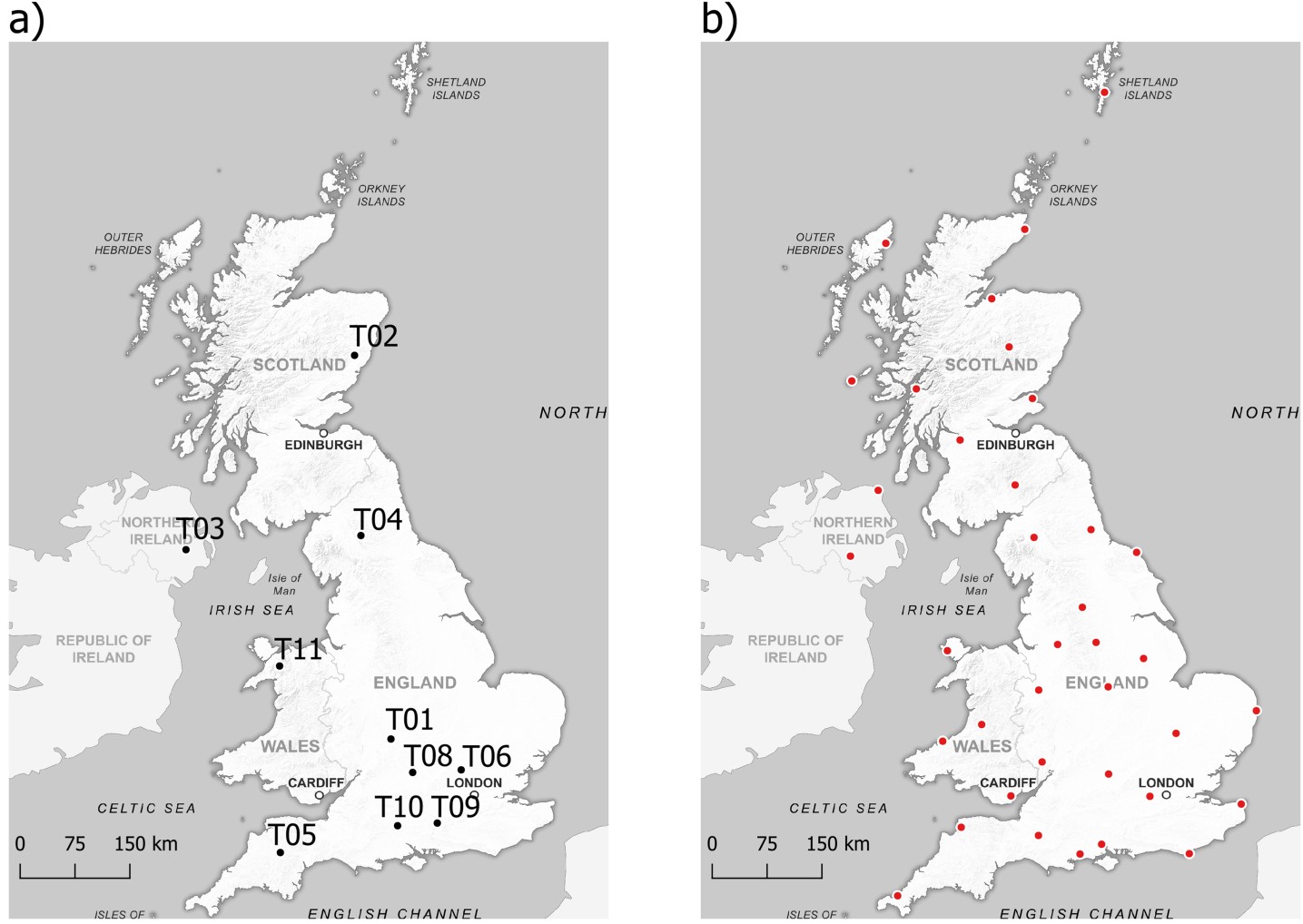

**Figure 1 Sampling locations of data used in the study.** (A) Locations of ponds sampled by the Environmental Change Network to obtain data on *R. temporaria* reproductive success and phenology, and agricultural pollutants, (B) Locations of weather stations the Met Office use to measure climatic variables. Background map source credit: GB Overview Maps, OGL. Data source credits: ECN site locations: https://catalogue.ceh.ac.uk/documents/4d8c7dd9-8248-46ca-b988-c1fc38e51581; Met Office site locations: https://www.metoffice.gov.uk/research/climate/maps-and-data/historic-station-data.

## METHODS

Portions of this text were previously published as part of an undergraduate thesis at the University of Exeter (*Oliver, 2023*).

### Data sources

We obtained data on *R. temporaria* reproductive success and phenology, as well as ammonium and nitrate ion concentrations, from the Environmental Change Network (ECN) (*Rennie et al., 2017*). Across the UK, there are 10 ECN sites which collect this data (Fig. 1A). The ECN sites differed in the number of ponds sampled and the duration they were sampled for Table 1.

From 1st January every year (see Table 1 for site-specific sampling coverage), ponds were sampled weekly until males first congregated at the pond. This date was recorded, and

**Table 1 The number of ponds per ECN site and the years each ECN site was sampled for.**

| ECN site | Latitude | Longitude | Number of ponds | First sampling year | Last sampling year |
|----------|----------|-----------|-----------------|---------------------|--------------------|
| T01 | 52.19361 | −1.76417 | 1 | 1994 | 2012 |
| T02 | 56.90917 | −2.55333 | 2 | 1994 | 2014 |
| T03 | 54.45333 | −6.07806 | 2 | 1994 | 2011 |
| T04 | 54.695 | −2.38778 | 7 | 1994 | 2015 |
| T05 | 50.78167 | −3.91778 | 1 | 1994 | 2015 |
| T06 | 51.80333 | −0.3725 | 1 | 1994 | 2010 |
| T08 | 51.781111 | −1.33583 | 1 | 1994 | 1994 |
| T09 | 51.15444 | −0.86306 | 1 | 1995 | 2001 |
| T10 | 51.12694 | −1.63972 | 1 | 2001 | 2015 |
| T11 | 53.07455 | −4.03351 | 2 | 1995 | 2012 |

subsequent sampling frequencies increased to daily until spawn first hatched. The dates of spawning and hatching were recorded. After hatching, sampling frequencies decreased to weekly until 16 weeks after spawning or when froglets were seen leaving the ponds. The date of leaving was recorded. The areas of spawn present and the percentage of spawn found dead were recorded when each pond was sampled.

The concentrations of ammonium and nitrate ions were measured from the date of spawning by taking 250 ml of pond water to analyse in a laboratory when each pond was sampled. We obtained data on land cover change from the UK Centre for Ecology & Hydrology (UKCEH) (*Rowland et al., 2020a*, *2020b*). Using satellite data, the dominant land cover type in each 25 × 25 m square of the UK in 1990 and 2015 were classified into six classes: woodland, arable, grassland, water, built-up areas, and other. We obtained data on climate change from 37 historic Met Office stations across the UK (Fig. 1B), which have been collecting climate data for at least 44 years (*Met Office, 2022*). Climate data consists of monthly mean daily maximum temperature (°C), mean daily minimum temperature (°C), days with air frost, and total rainfall (mm).

## Data processing

To quantify the land cover change surrounding each ECN location and the distances between each ECN location and each Met Office station, we used QGIS (*QGIS Development Team, 2022*). Coordinates for the ECN locations were provided by the ECN, and the coordinates for the Met Office stations were taken from the historic station data webpage (*Met Office, 2022*), using the coordinate reference system OSGB 1996/British National Grid. To calculate the land cover change surrounding each ECN location we created a buffer zone with a radius of 2.5 km surrounding each ECN location, as some common frogs can disperse over 2 km (*Kovar et al., 2009*). We then used the land cover data obtained from UKCEH to calculate the area of each land cover class within these buffers in both 1990 and 2015. To calculate the distances between each ECN location and each Met Office station, we used the distance matrix tool.

**Table 2  The variables extracted for analysis and their sources.**

| Extracted variable | Abbreviation | Unit of measurement | Data source |
|---|---|---|---|
| Congregation date | cong_date | Julian days | *Rennie et al. (2017)* |
| Spawning date | spawn_date | Julian days | *Rennie et al. (2017)* |
| Hatching date | hatch_date | Julian days | *Rennie et al. (2017)* |
| Leaving date | leave_date | Julian days | *Rennie et al. (2017)* |
| Surface area of spawn | surf_area | m² | *Rennie et al. (2017)* |
| Percentage of spawn dead | perc_dead | % | *Rennie et al. (2017)* |
| Ammonium concentration | spawn_nh4n | mg/l | *Rennie et al. (2017)* |
| Nitrate concentration | spawn_no3n | mg/l | *Rennie et al. (2017)* |
| Woodland area | woodland_area | m² | *Rowland et al. (2020a, 2020b)* |
| Arable area | arable_area | m² | *Rowland et al. (2020a, 2020b)* |
| Grassland area | grassland_area | m² | *Rowland et al. (2020a, 2020b)* |
| Freshwater body area | freshwater_area | m² | *Rowland et al. (2020a, 2020b)* |
| Built-up area | urban_area | m² | *Rowland et al. (2020a, 2020b)* |
| Other land cover area | other_area | m² | *Rowland et al. (2020a, 2020b)* |
| Mean daily maximum temperature per month | tmax | °C | *Met Office (2022)* |
| Mean daily minimum temperature per month | tmin | °C | *Met Office (2022)* |
| Number of days with air frost per month | af | Days | *Met Office (2022)* |
| Total rainfall per month | Rain | mm | *Met Office (2022)* |

We used the software *R* (*R Core Team, 2020*) to create a dataframe for each reproductive success and phenological response variable measured by the ECN (Table 2). Each row represented a pond at an ECN location in a single year from 1994–2015. The reproductive success data frames contained the largest surface area of spawn and percentage of dead spawn at each pond in each year. The phenology data frames contained the earliest date of either congregation, spawning, hatching, or leaving at each pond in each year, all converted to days since the 1$^{st}$ January (DOY).

We incorporated the data on maximum ammonium and nitrate ion concentrations recorded in each pond in each year whilst breeding adults were present into the spawning dataset. We added data on mean ammonium and nitrate ion concentrations recorded in each pond in each year whilst spawn was present to the data frames on the surface area of spawn, the percentage of spawn dead, and the date of hatching.

To calculate land cover change, we made assumption that the *rate* of land cover change was constant between years, allowing us to estimate the area of each land cover class surrounding each ECN location in each year from 1994–2015. We identified the nearest Met Office station to each ECN location and used the climate data from these stations as estimates for the climate at the ECN locations. As congregation and spawning dates depend on winter climate (*Carroll et al., 2009*; *Benard, 2015*), we calculated the average of each climatic variable at each ECN location in each year over the winter months (December–February). This yielded a single mean temperature datum for that 3 month period per pond per year. Similarly we calculated the average of each climatic variable at

each ECN location in each year during the months whilst spawn was present (January–May), giving a single datum for that 5 month period per pond per year. We added these values to the data frames containing information on the surface area of spawn, the percentage of spawn dead, and the date of hatching.

## Data analysis

All data and code to reproduce these analyses is provided at https://github.com/xavharrison/FrogSpawn2023. First, for all variables (spawning, hatching, and percentage dead) we plotted temporal trends from 1995 to 2015 at the site level for the seven sites with sufficient time series. We used Pearson correlations to identify significant temporal trends in these traits after correcting for multiple testing. Sample sizes for the temporal analysis were: spawning ($n = 175$); hatching ($n = 145$); percentage dead spawn ($n = 178$). Then, we used bivariate response mixed effects models on the same dataset to test for correlations between spring temperature (spawning) or spawning temperature (hatching, percentage dead) and phenological variables. Bivariate mixed effect models can estimate the posterior correlation between temperature and phenological outcome variables whilst controlling for site effects, temporal trends and temporal autocorrelation (*Houslay & Wilson, 2017*; *Harrison, 2021*). This approach is similar to 'detrending' residuals to identify causal effects (*e.g.*, see *Votier et al., 2008*). By measuring the correlation among the *residuals* of the two responses (*e.g.*, mean maximum spawning temperature and hatch date), we can identify shifts in timing or changes to mortality over and above any long term trends in advancement of breeding. Here we predict that higher than average maximum temperatures (a positive residual) would be associated with earlier than average hatch dates (a negative residual). This manifests as a negative correlation in the residuals of the model, assesses as significant/important depending on whether the credible intervals cross zero.

Finally, we used generalised linear models (GLMs) and general linear mixed effects models (GLMM) to investigate drivers in variation in date of spawning ($n = 23$), and hatching ($n = 25$), as well as the percentage of dead spawn ($n = 33$) at the landscape scale. Variation in the size of the datasets is a result of ensuring all rows have complete information on all traits, including land use change and pollution. Landscape-scale variables such as land cover, temperature and chemical use are often correlated, so we used a principal components analysis (PCA) on the anthropogenic stressor variables for each of the reproductive success and phenology data frames using the factoextra and FactoMineR packages (*Sebastien Le & Husson, 2008*; *Mundt, 2020*). We extracted Principal Component 1 (PC1) and Principal Component 2 (PC2) for each reproductive success and phenological variable to use as predictors in our modelling. We used rotation plots to identify the anthropogenic stressors explained by each principal component (*Wei & Simko, 2021*), and so crucially the biological interpretation of the importance of each PC changes for each model. For each outcome variable we used Spearman's rank correlation to detect significant associations with the two primary axes of variation in the PCA (PC1/PC2).

## RESULTS

### Spawn date

Across seven sites, only T01 showed a significant negative relationship with time, indicative of advancing timing of spawning (cor = −0.7, p.adj = 0.01 Fig. 2A). Here, spawning is advancing at approximately 14.9 days per decade. Sites T02 and T11 also exhibited moderate negative correlations (−0.42 and −0.3 respectively) but were not significant after correction for multiple testing. A bivariate response model controlling for temporal autocorrelation recovered a site-level effect of timing of spawning, where sites with higher winter average temperatures tend to initiate spawning earlier (posterior correlation −0.55%, 95% credible interval −0.82, −0.1; Fig. 2B; Table 3A). However we found no evidence of a residual correlation between winter temperature and spawning date at the datum level (Table 3A), suggesting limited plasticity in spawn date *i.e.*, warmer than average winters do not consistently result in earlier than average spawning.

Univariate correlations with corrections for multiple testing revealed negative relationships between spawn date and both arable area and mean maximum winter temperature (higher values mean earlier spawning), and positive relationships with air frost days and grassland area (Figs. 2C–2F. Table S1A; higher values mean later spawning). Principal components analysis identified that the majority of these variables were collinear (Table S2; Fig. S1A), and recovered a positive correlation between PC1 and spawn date (cor = −0.45, $p$ = 0.03).

### Hatch date

Site T04 (Upper Teesdale, North Pennines) exhibited a negative trend over time in hatch date, indicative of earlier hatching every year ($p$ = −0.4, p.adj = 0.045; Fig. 3A). Sites T01, T02 and T011 also showed negative trends (correlations −0.23 to −0.44), though these were not significant after correction for multiple testing. A bivariate response model controlling for temporal autocorrelation recovered a negative correlation between hatch date and mean maximum spawning temperature at the site level (*i.e.*, sites with higher spawning temperatures tend to hatch earlier, though the credible intervals crossed zero (posterior correlation −0.46%, 95% credible interval −0.83, 0.03; Fig. 2B; Table 3B). We did find evidence of a residual correlation between spawning temperature and hatch date at the datum level (Table 3B), suggesting earlier hatching in warmer years consistent with faster development of eggs.

Univariate correlations with corrections for multiple testing revealed negative relationships between hatch date and both arable area and mean maximum spawning temperature (higher values mean earlier hatching), a positive, potentially non-linear relationship with air frost days, and a positive relationship with grassland area (Figs. 3C–3F. Table S1A; higher values mean later hatching). Principal components analysis identified that the majority of these variables were collinear (Table S3; Fig. S1B). Increaases in PC1 (representing fewer air frost days, higher temperatures and greater amounts of arable area) resulted in earlier hatching (cor = −0.7, $p$ < 0.001).

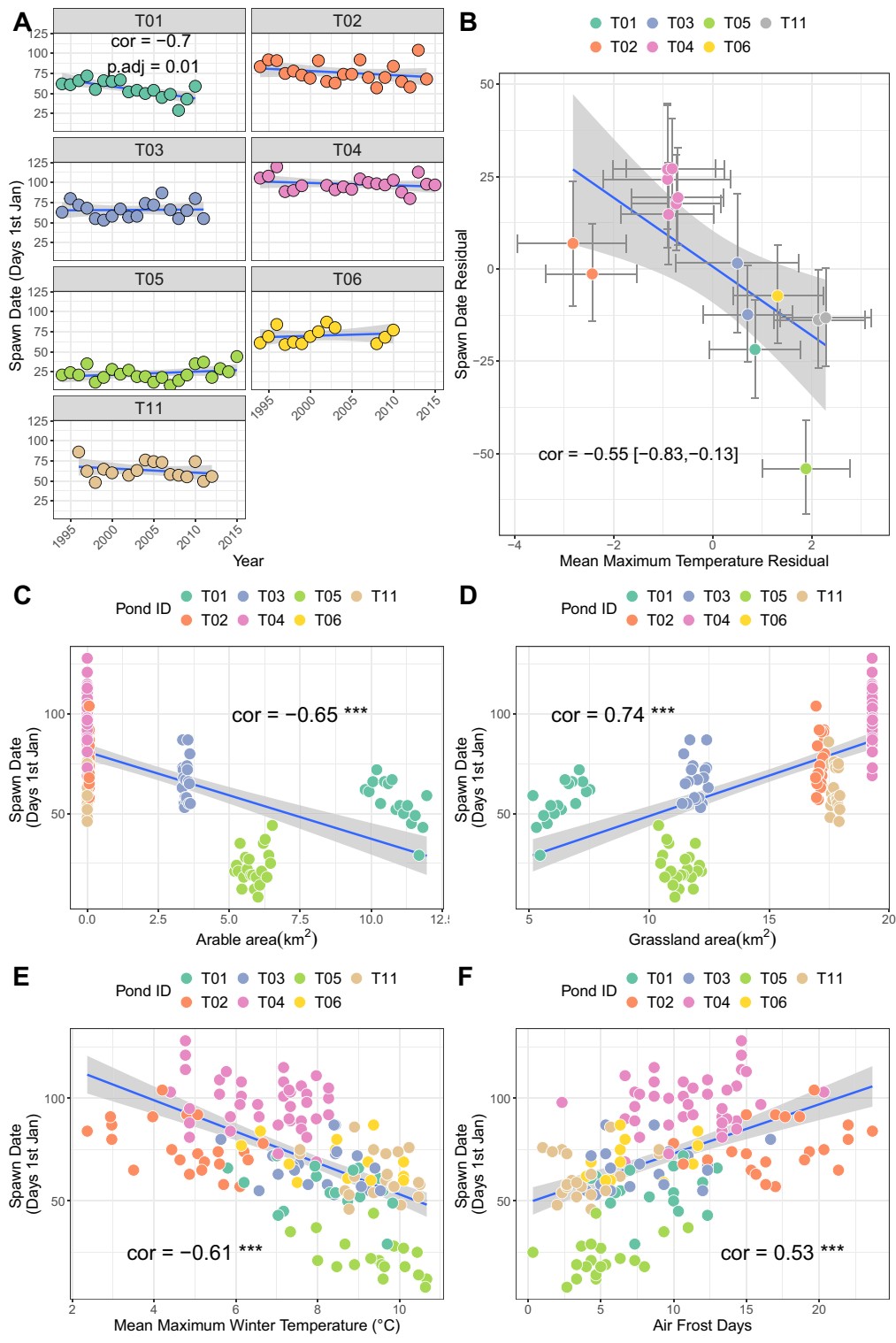

**Figure 2 Factors associated with differences in spawning date.** (A) Trends in spawning date over time for seven sites. (B) Results from bivariate mixed effects model examining posterior correlation between spawn date and mean maximum winter temperature. (C) Correlation between spawn date and arable area. (D) Correlation between spawn date and grassland area. (E) Correlation between spawn date and mean maximum winter temperature. (F) Correlation between spawn date and number of air frost days.

![PeerJ]

**Table 3 Bivariate mixed effects model estimates.** (A) Spawn date, (B) hatch date and (C) percentage of dead spawn. The bold values indicate parameters for which the credible intervals do not cross zero.

**(A) Spawn date**

Correlation structures:

| Variable | Estimate | Error | Lower 95% | Upper 95% |
| --- | --- | --- | --- | --- |
| AR spawn | 0.47 | 0.08 | 0.32 | 0.62 |
| AR Tmax | 0.82 | 0.05 | 0.73 | 0.91 |

Pond ID ($n = 15$)

| Variable | Estimate | Error | Lower 95% | Upper 95% |
| --- | --- | --- | --- | --- |
| sd (Spawn) | 23.85 | 4.8 | 16.41 | 35.36 |
| sd (Tmax) | 1.69 | 0.33 | 1.17 | 2.48 |
| **cor (Spawn, Tmax)** | **−0.55** | **0.19** | **−0.84** | **−0.1** |

Regression coefficients:

| Variable | Estimate | Error | lower 95% | upper 95% |
| --- | --- | --- | --- | --- |
| spawn_Intercept | 631.85 | 484.78 | −320.36 | 1,608.35 |
| tmax_Intercept | 9.42 | 83.91 | −159.81 | 172.67 |
| spawn_year | −0.28 | 0.24 | −0.76 | 0.2 |
| spawn_tmax | 0 | 0.04 | −0.08 | 0.08 |

| Variable | Estimate | Error | Lower 95% | Upper 95% |
| --- | --- | --- | --- | --- |
| sigma_spawn | 9.51 | 0.54 | 8.52 | 10.6 |
| sigma_tmax | 0.64 | 0.04 | 0.57 | 0.72 |

Residual correlations:

| Variable | Estimate | Error | Lower 95% | Upper 95% |
| --- | --- | --- | --- | --- |
| rescor (spawn, tmax) | −0.1 | 0.08 | −0.26 | 0.06 |

**(B) Hatch date**

Correlation structures:

| Variable | Estimate | Error | Lower 95% | Upper 95% |
| --- | --- | --- | --- | --- |
| AR Hatch | 0.34 | 0.08 | 0.17 | 0.5 |
| AR Tmax | 0.66 | 0.06 | 0.54 | 0.79 |

Pond ID ($n = 14$)

| Variable | Estimate | Error | Lower 95% | Upper 95% |
| --- | --- | --- | --- | --- |
| sd (Hatch) | 22.32 | 4.75 | 15.12 | 33.52 |
| sd (Tmax) | 1.48 | 0.31 | 1.01 | 2.21 |
| cor (Hatch, Tmax) | −0.46 | 0.21 | −0.8 | 0.03 |

Regression coefficients:

| Variable | Estimate | Error | Lower 95% | Upper 95% |
| --- | --- | --- | --- | --- |
| hatch_Intercept | 754.4 | 443.85 | −141.41 | 1,613.06 |
| tmax_Intercept | −45.18 | 50.58 | −152.78 | 49.28 |
| hatch_year | −0.33 | 0.22 | −0.76 | 0.12 |
| tmax_year | 0.03 | 0.03 | −0.02 | 0.08 |

| Variable | Estimate | Error | Lower 95% | Upper 95% |
| --- | --- | --- | --- | --- |
| sigma_hatch | 9.83 | 0.64 | 8.66 | 11.19 |
| sigma_tmax | 0.6 | 0.04 | 0.53 | 0.68 |

*(Continued)*
| Table 3 (continued) | | | | |
|---|---|---|---|---|
| **(B) Hatch date** | | | | |
| Residual correlations: | | | | |
| **Variable** | **Estimate** | **Error** | **Lower 95%** | **Upper 95%** |
| **rescor (Hatch, Tmax)** | **−0.37** | **0.08** | **−0.52** | **−0.21** |
| **(C) Percent dead spawn** | | | | |
| Correlation structures: | | | | |
| **Variable** | **Estimate** | **Error** | **Lower 95%** | **Upper 95%** |
| AR PercentDead | 0.02 | 0.09 | −0.15 | 0.19 |
| AR Tmax | 0.82 | 0.06 | 0.71 | 0.93 |
| Pond ID (*n* = 11) | | | | |
| **Variable** | **Estimate** | **Error** | **Lower 95%** | **Upper 95%** |
| sd (PercentDead) | 5.22 | 1.18 | 3.44 | 8.15 |
| sd (Tmax) | 1.29 | 0.29 | 0.87 | 1.99 |
| **cor (PercentDead, Tmax)** | **−0.8** | **0.13** | **−0.96** | **−0.47** |
| Regression coefficients: | | | | |
| **Variable** | **Estimate** | **Error** | **Lower 95%** | **Upper 95%** |
| PercentDead_Intercept | −76.58 | 146.72 | −370.84 | 208.55 |
| Tmax_Intercept | −61.13 | 72.79 | −212.43 | 72.37 |
| PercentDead_year | 0.04 | 0.07 | −0.11 | 0.18 |
| Tmax_year | 0.04 | 0.04 | −0.03 | 0.11 |
| **Variable** | **Estimate** | **Error** | **Lower 95%** | **Upper 95%** |
| sigma_PercentDead | 4.61 | 0.26 | 4.14 | 5.16 |
| sigma_Tmax | 0.54 | 0.03 | 0.48 | 0.61 |
| Residual correlations: | | | | |
| **Variable** | **Estimate** | **Error** | **Lower 95%** | **Upper 95%** |
| rescor (PercentDead, Tmax) | 0.04 | 0.08 | −0.12 | 0.21 |

## Reproductive success

Though several sites showed increases over time in the percentage of dead spawn, only site T11 (Yr Wyddfa, Wales) had a significant trend when correcting for multiple testing (cor = 0.54, p.adj = 0.02; Fig. 4A). Bivariate response modelling identified a negative posterior correlation between temperature during spawning and spawn death (cor = −0.8%, 95% credible interval −0.96, −0.45; Table 3C) suggesting that sites with lower maximum spawning temperatures tend to show greater degrees of spawn mortality, on average. We did not detect correlation between the residuals for spawn death and spawn temperature (posterior mean correlation 0.04%, 95% credible intervals −0.12, 0.21), meaning that within sites there was no evidence that years with colder than average spawning temperatures suffered higher than average mortality. Univariate analyses revealed a negative relationship between arable area and percentage of dead spawn (more arable area reflects lower mortality), and a positive relationship with grassland area (Figs. 4C–4E. Table S1C). Principal components analysis identified strong collinearity among

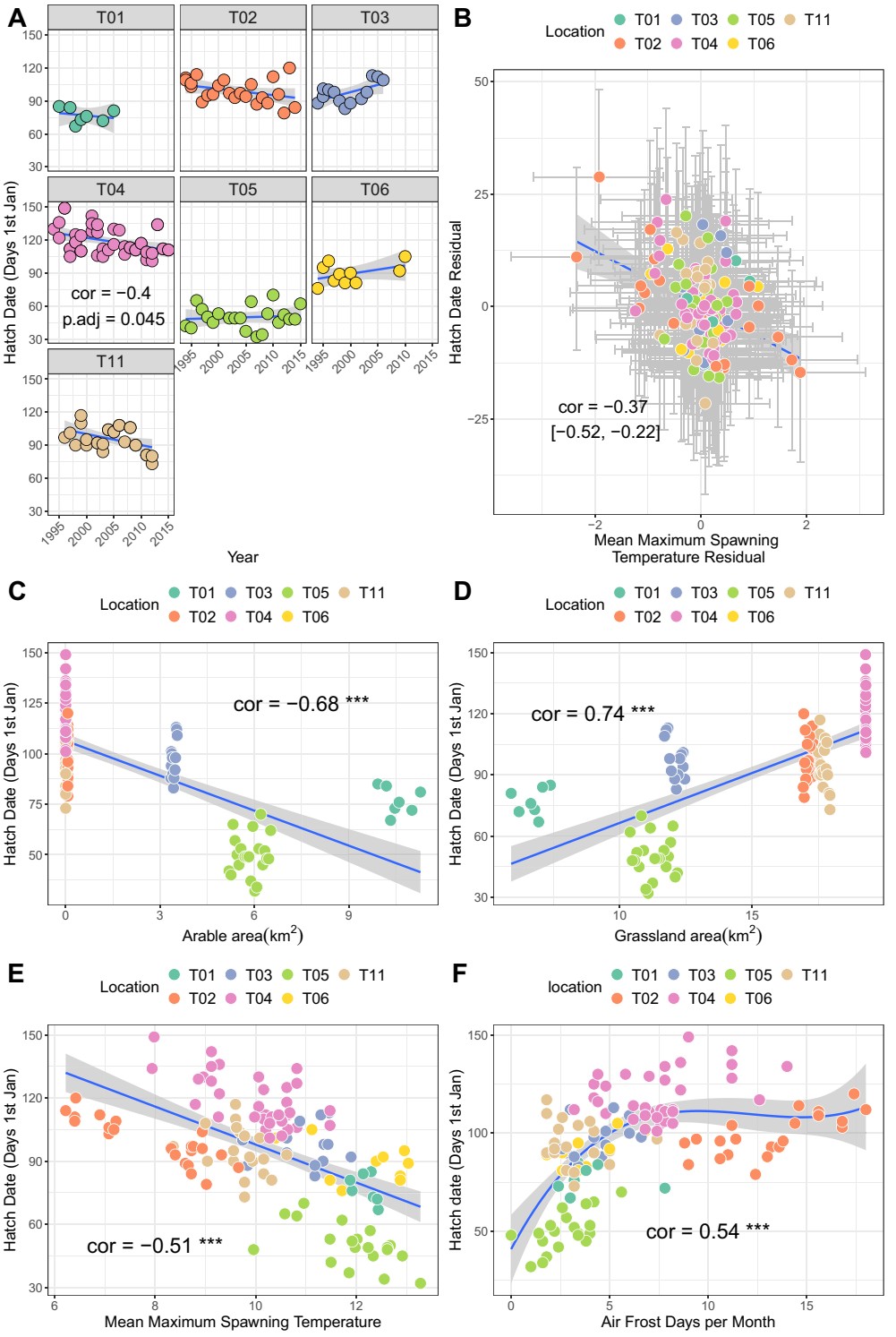

**Figure 3** **Factors associated with variation in hatch date.** (A) Trends in hatch date over time for seven sites. (B) Results from bivariate mixed effects model examining posterior correlation between hatch date and mean maximum winter temperature. (C) Correlation between hatch date and arable area. (D) Correlation between hatch date and grassland area. (E) Correlation between hatch date and mean maximum spawning temperature. (F) Correlation between hatch date and number of air frost days.

these traits (Table S4; Fig. S1C). Increases in PC1 (representing fewer air frost days, higher mean temperatures and more arable area. Table S4) resulted in lower proportions of dead spawn (cor = 0.49, $p$ = 0.007).

## DISCUSSION

Here, we used long-term data on the phenology and reproductive success of UK *Rana temporaria* to identify associations with climate, land use change, and pollution. We uncovered site-specific temporal trends in both date of spawning and hatching, suggestive of advancing phenology in some but not all areas of the UK. Moreover, we found a signal of increased spawn mortality over time at some sites, suggesting the drivers of spawn mortality are not uniform at the landscape scale. In addition to temperature, all reproductive traits we measured could also be linked to traits of land use change and pollution (*via* univariate tests and principal component analysis), which themselves correlated with temperature variation. This study highlights the important role of winter and spawning temperature in driving variation in amphibian breeding phenology, but also the complexity of disentangling the relative significance of multiple correlated environmental variables in wild phenology studies.

### Phenology, land use change and temperature

This work is consistent with previous studies linking temperature shifts to alteration in amphibian breeding phenology (*e.g.*, *Carroll et al., 2009*; *Phillimore et al., 2010*; *Blaustein et al., 2010*; *Benard, 2015*). Consistent with our predictions, at the landscape scale we found that variation in the onset of spawning and hatching was linked to warmer average winter and spawning temperatures, respectively. Similarly, *within* populations we found that the magnitude of the slope of phenological advancement was population specific, as did a previous study on *R temporaria* (*Phillimore et al., 2010*). Spatial heterogeneity in the slope of phenological trends may be common (*e.g.*, *Primack et al., 2009*), and not solely associated with latitudinal variation. For example, *Sparks et al. (2007)* found considerable variation in the slope of shifts in spawning initiation dates with mean Jan-March temperature between populations of *R temporaria* in the UK and Poland, even though they were found at similar latitudes. Both studies (*Sparks et al., 2007*; *Phillimore et al., 2010*) attribute the observed among-population variation in slope to local adaptation to climate. Spatial variation in the strength of phenological (*Primack et al., 2009*; *Phillimore et al., 2010*; this study) or phenotypic (*Sheridan et al., 2018*) adjustments to climate change highlight that 'space-for-time' studies of species traits under climate change should be used with caution (*Phillimore et al., 2010*; *Sheridan et al., 2018*).

We still lack a comprehensive understanding of the costs of a population not 'keeping up' with rates of phenological advancement, nor how this may vary in concert with the mean and variance of local climatic conditions. For example, warmer winters have been shown to advance phenology in congeners like wood frogs (*Rana sylvatica*), but also delay larval development time when followed by cooler post-breeding temperatures (*Benard, 2015*). Thus, even if environmental trait means shift in a consistent direction (winters get warmer, earlier), increased variation around that mean may erode selection on changes in

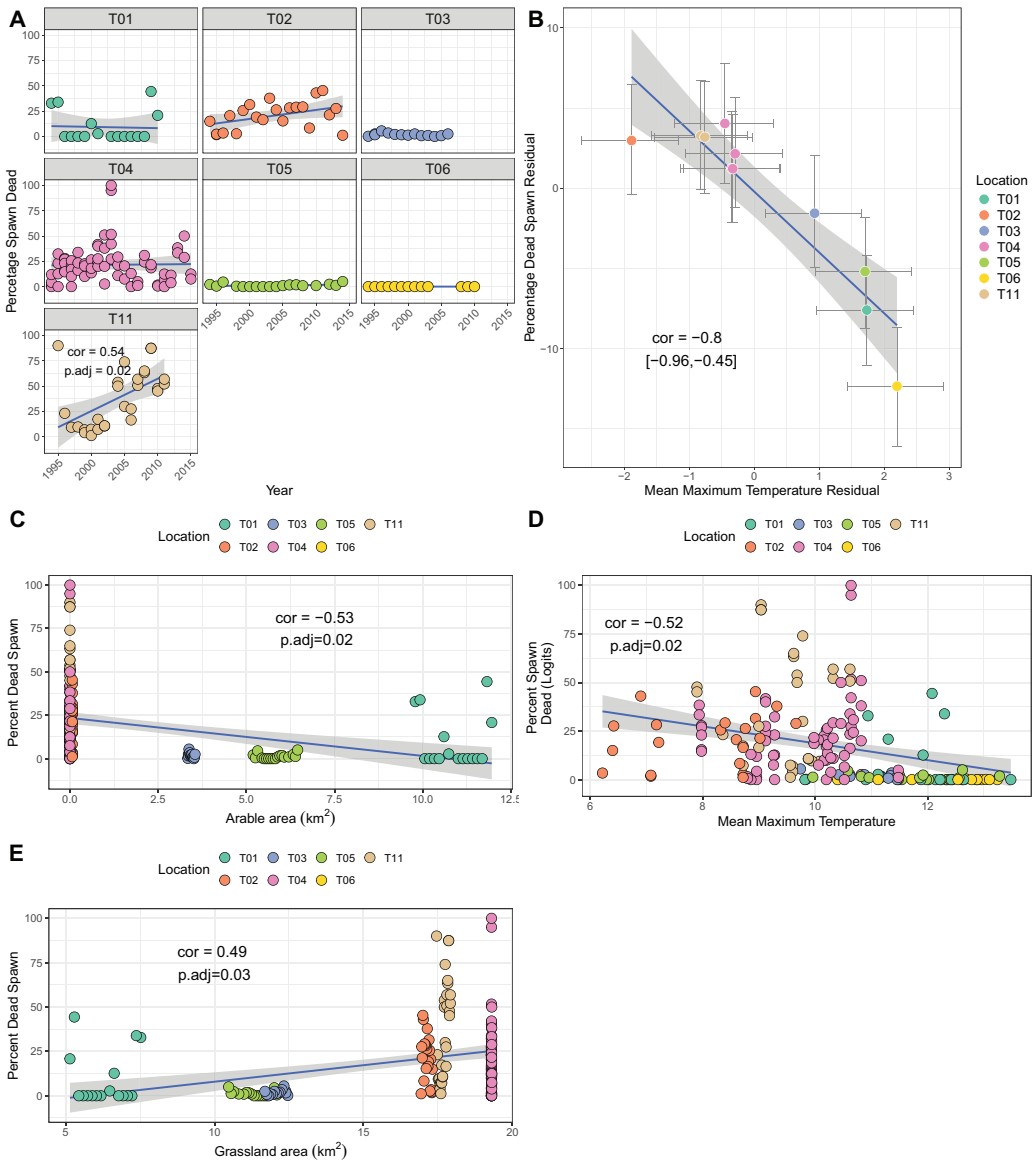

**Figure 4 Factors associated with the percentage of dead spawn.** (A) Trends in percentage of dead spawn over time for seven sites. (B) Results from bivariate mixed effects model examining posterior correlation between percentage of dead spawn and mean maximum spawning temperature. (C) Correlation between percentage of dead spawn and arable area. (D) Correlation between percentage of dead spawn and mean maximum spawning temperature. (E) Correlation between spawn date and grassland area.

phenological traits. In common toads *Bufo bufo* in the United Kingdom, warmer than average temperatures correlated with decreased body condition and survival (*Reading, 2007*). Recent work on wood frogs showed that phenological shifts can expose individuals to colder temperatures and resulted in lower tolerance of offspring to pollutants like NaCl (*Buss, Swierk & Hua, 2021*). Collectively these data suggest that in some species, warming can induce cryptic cost of breeding plasticity in multiple life stages that are not immediately apparent if looking at phenological variables alone (see *Blaustein et al., 2010*).

Using bivariate models revealed that within sites, warmer than average winter temperatures for a particular year were not associated with earlier than average spawning, indicating limited *short-term* (rather than inter-annual) plasticity in the timing of spawning. However, we did uncover such a pattern for hatching, consistent with more rapid development times under warmer spawning temperatures. Previous work on natterjack toads has shown that devleopment time is reduced under warming temperatures (*Sanuy, Oromí & Galofré, 2008*), similar to the plasiticty we have observed here. The lack of signal of short-term plasticity in spawning could arise if mean maximum winter temperature is too coarse a measurement to capture the climatic cues used within years to determine exact spawning date. Conversely, such plasticity might be the property of individuals and not populations. For example, previous work on two terrestrial newt species found marked among-individual variation in exploratory behaviour linked to temperature shifts (*Hubáček & Gvoždík, 2023*). This suggests individual frogs may be capable of adjusting their reproductive behaviour in response to fine-scale among-year variation in climate (plasticity), but against a background of strong among-individual heterogeneity, making the pattern difficult to detect at the population level. Addressing this possibility requires that we track individual frogs across years, for example by using unique dorsal markings (*Petrovan et al., 2024*), to measure individual behavioural adjustments over time.

We also uncovered associations between land use change and breeding phenology. Increased arable and built-up land cover is associated with earlier hatching, whilst grassland is associated with later hatching. Arable and built-up areas tend to be warmer than the surrounding natural habitats due to unvegetated ground, whilst grassland areas, with higher vegetation levels, are cooler (*Lembrechts, Nijs & Lenoir, 2019*; *Schmidt, Lischeid & Nendel, 2019*). The warming effects of arable and built-up areas can even increase the temperature of the wider landscape, leading to earlier breeding phenology in the areas surrounding human-dominated land (*Tian et al., 2020*). These effects could be reduced by increasing vegetated areas in these land cover types, thus providing cooler microclimates (*Greenwood et al., 2016*). Previous work has shown that anthropogenic habitat alteration can alter breeding phenology in multiple Australian amphibian species, specifically that increased urbanisation is linked to earlier breeding (*Liu et al., 2022*). However, the nature of our data prevents us from disentangling the relative influence of land use change and temperature regime shifts. Future work could investigate microclimatic variation at breeding sites to explore how urbanisation changes climatic envelope experienced by amphibians, and compare these trends to sites with similar mean temperatures in (more southerly) rural areas.

## Spawn mortality, land use change and temperature

We found no clear association between mean winter maximum or minimum temperatures and spawn mortality, when temperature was used as the sole predictor in models. Instead, using Principal Component Analysis we found that the percentage of dead spawn was linked to a composite measure of frost days, rainfall, land use change, and temperature. Increases in the proportion of arable and urban areas, nitrate levels, and increased mean

maximum temperature, were associated with higher spawn survival. Conversely, increased numbers of frost days, rainfall and grassland areas were associated with decreasing reproductive success. A key finding was that some populations are exhibiting consistent increases in the proportion of dead spawn over time, which may indicate that long term viability of these locations may be compromised even if no further land use or environmental changes occur.

The correlations between land use, pollution and spawn survival did not align with our predictions. We expected increased farmland (and associated pollution such as nitrates), and increased urbanisation to be detrimental rather than associated with higher spawn survival. However, though human-driven land use change is often associated with lower reproductive success, some studies have shown that they can support declining populations. For example, the average reproductive success of multiple amphibian species in America breeding in arable ponds was no different to in natural wetlands (*Knutson et al., 2004*). Though *some* species did respond negatively to arable land use (*Knutson et al., 2004*), *R. temporaria* have been found to use arable ponds more than other amphibians (*Hartel, Băncilă & Cogălniceanu, 2011*). Therefore, arable land cover may be able to support robust populations of *R. temporaria*, leading to higher spawn survival in the ECN sampling locations. Arable ditches may confer landscape connectivity for amphibians, reptiles, and mammals; *Maes, Musters & De Snoo (2008)* found that ditches in agricultural environmental schemes supported similar *Rana esculenta* abundances as nature reserves, demonstrating how these features of arable land cover can serve as important avenues for dispersal between breeding sites. This is crucial in preventing population isolation, perhaps explaining why we found a negative relationship between the arable area and the maximum percentage of spawn that died (*Allentoft & O'Brien, 2010*).

Built-up areas have also been shown to increase the fitness of some wildlife populations, with some threatened birds (*Kettel et al., 2019*) and amphibians (*Iglesias-Carrasco, Martín & Cabido, 2017*) having greater reproductive success ad body condition respectively in built-up areas than in the countryside. There are multiple theories for why this could be *Saenz, Hall & Kwiatkowski (2015)* suggest that the occurrence of chytridiomycosis, an amphibian disease that can reduce fitness, could be lower in urban areas. However, chytridiomycosis is not common in British amphibians (*Garner et al., 2005*) so this is unlikely to explain our findings. *Hall & Warner (2017)* suggest that the high densities of prey insects in urban areas or lower predation pressures that allow adults to spend more time hunting, could increase fitness. Further evidence for this is from Germany, where *R. temporaria* adults were found to be bigger in urban greenspace than in the surrounding countryside (*Niemeier et al., 2020*). Larger adult body size is likely to increase offspring survival (*Hall & Warner, 2017*), providing an explanation as to why built-up areas are associated with lower *R. temporaria* spawn mortality.

Many of the studies that demonstrate the value of human-dominated land cover types also acknowledge the need for management. In built-up areas, it is important to reduce barriers to movement by connecting urban greenspace (*Mazgajska & Mazgajski, 2020*; *Niemeier et al., 2020*). These suggestions highlight that local management for

*R. temporaria* could be more important that the broad-scale land cover type in determining this amphibian's reproductive success. Landscape management practices could also explain why grassland is negatively associated with *R. temporaria* reproductive success (*i.e.*, positively associated with mortality). In the UK, only 2% of grassland is classed as diverse (*Bullock et al., 2011*), due to widespread "improvement" (*Vickery et al., 1999*) and livestock grazing (*Fuller, 1987*; *Bullock et al., 2011*). However, grazing can be detrimental to amphibians. Livestock can cause high levels of wetland bank erosion (*Trimble, 1994*), leading to increased sediment deposition. When investigating the impact of cattle on *Bufo achalensis*, a toad species endemic to Argentina, *Jofré, Reading & di Tada (2007)* found that increased sediment levels reduced algal growth, a key food source for larval amphibians, leading to higher mortality in the *B. achalensis* larvae. This could result in increased isolation between *R. temporaria* populations, potentially leading to the increased percentage of spawn death observed in ECN locations surrounded by grassland. High densities of livestock can also directly reduce amphibian spawn reproductive success through disruption and trampling (*Knutson et al., 2004*).

Our modelling could not conclusively implicate temperature regime changes as a potential driver of difference in amphibian reproductive success, though maximum temperature and number of frost days were part of the composite measure (PC1) associated with spawn survival. Although climate change is leading to warmer temperatures, it is also causing extreme weather conditions (*Huber & Gulledge, 2011*). One example is the occurrence of spring cold-snaps, a phenomenon that has been shown to have detrimental effects on a wide range of organisms (*Augspurger, 2013*; *Benard, 2015*; *Turner & Maclean, 2022*). *Benard (2015)* observed an increase in *Rana sylvatica* larvae being exposed to cold-snaps from 2006 to 2012, leading to altered development. Freezing is known to kill *R. temporaria* (*Pasanen & Karhapää, 1997*), making it likely that cold-snaps could lead to high spawn mortality. Warmer springs and fewer frost days likely explain the lower proportions of dead spawn observed in this dataset under these conditions.

## CONCLUSIONS AND FUTURE WORK

Here, we have demonstrated associations between climate, land use change and parameters of amphibian breeding success and phenology at the landscape scale. We found marked variation among populations in the magnitude of change over time in spawning date, hatching date and spawn mortality. These patterns suggest that not all populations of common frog respond to environmental stressors, or environmental change, in the same way. These data are useful for understanding which populations may be most at risk, but without understanding the drivers of such patterns we lack the ability to make predictions for new populations and prioritise conservation efforts accordingly. Future work on *R. temporaria* in the UK should prioritise pond-scale approaches to investigations of the factors shaping phenology and reproductive success, which will permit measurement of microhabitats experienced by breeding adults. Microhabitat temperature measurements could shed further light on the frequency and consequences of freezing temperature (*i.e.*, lower winter minima) on *R. temporaria*, as well as their effect on larval development and survival. Similarly, fine-scale land-use data could aid in identifying the habitats most

beneficial for successful *R. temporaria* breeding, allowing the efficacy of management practices to be optimised. The need for fine-scale data is particularly important due to the small size and limited dispersal ability of *R. temporaria* (*Kovar et al., 2009*), and thus may also be applicable to other such species. For the ongoing survival of *R. temporaria* it is vital to reduce the harm of extreme weather events due to climate change, which may be achieved through the management of microclimates. Afforestation and re-flooding drained wetlands, for example, could help maintain stable and favourable microclimatic envelopes. These practices could also aid *R. temporaria* by increasing vegetation complexity and providing additional breeding locations and their connectivity, both of which could improve *R. temporaria* survival.

### Funding
The authors received no funding for this work.

### Competing Interests
Xavier Harrison is an Academic Editor for PeerJ.

### Author Contributions
- Kat E. Oliver conceived and designed the experiments, performed the experiments, analyzed the data, prepared figures and/or tables, authored or reviewed drafts of the article, and approved the final draft.
- Xavier A. Harrison conceived and designed the experiments, analyzed the data, prepared figures and/or tables, authored or reviewed drafts of the article, and approved the final draft.

### Data Availability
All data and code required to reproduce these analyses are available at GitHub: https://github.com/xavharrison/FrogSpawn2023

Raw data from the ECN are available in *Rennie et al. (2017)*.

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
