# Peer review of "Temperature and land use change are associated with Rana temporaria reproductive success and phenology"

_PeerJ, doi:10.7717/peerj.17901_

## Round 0.1 · original submission · Major Revisions

Please pay attention to comments from Reviewer 1. In particular, the paper seems in need of some major revisions to the text, including (1) a more complete review of the literature, (2) simplification of results, and (3) overall editing for clarity

·

Basic reporting

The manuscript has some important deficiencies that must be corrected in order to be accepted.
First, I find lacking a total absence of the data on which work has been done. I think in Suppl. Matt. or in some repository, tables with data on land use, phenology, years and water analysis must be included.
In addition, throughout the reading the manuscript at no time are we told what dates we are dealing with. Really are spring temperatures? I don’t know. Nor can we see the level of change in the start dates of the breeding period, spawn, land uses or the values of the variables used. Basic data are missed.
Secondly, the presentation of the results is somewhat cryptic and difficult to understand. There is no factor score tables from any of the principal component analysis. Perhaps correspondence analysis could be more visual. Authors don’t include any significance levels of the estimated correlations or models given.
Thirdly, there is a great deficiency in the bibliography consulted. There is a lot of work that needs to be included in the introduction and especially in the discussion.

Experimental design

In this manuscript, the authors analyse the influence of temperature, in relation to climate change, land cover change and chemical pollution on the breeding success of the Common frog (Rana temporaria).
The objective of this work is interesting since it proposes the analysis of the breeding period from different perspectives, or in other words, analyzing the influence of different anthropic changes on the reproduction of the Common frog.
Most studies use day of the year (DOY) as a variable to test how the start of breeding success varies in different years and to analyses its association with climatic variables. Only in Suppl. Mat. Use this variable but it isn’t explained I methodology.
I believe that in its present form the manuscript cannot be accepted. It needs a very deep restructuring and a greater and clearer development of the analysis of the variables used. The complete lack of tables with raw data makes it impossible to verify how the analyzes have been carried out. It is also not detailed how the models were developed and when the predictor variables were used.

Validity of the findings

The objective of this work is interesting since it proposes the analysis of the breeding period from different perspectives, or in other words, analyzing the influence of different anthropic changes on the reproduction of the Common frog.
However, the manuscript has some important deficiencies that must be corrected in order to be accepted.

Additional comments

In this manuscript, the authors analyse the influence of temperature, in relation to climate change, land cover change and chemical pollution on the breeding success of the Common frog (Rana temporaria).
The objective of this work is interesting since it proposes the analysis of the breeding period from different perspectives, or in other words, analyzing the influence of different anthropic changes on the reproduction of the Common frog.
However, the manuscript has some important deficiencies that must be corrected in order to be accepted.
First, I find lacking a total absence of the data on which work has been done. I think in Suppl. Matt. or in some repository, tables with data on land use, phenology, years and water analysis must be included.
In addition, throughout the reading the manuscript at no time are we told what dates we are dealing with. Really are spring temperatures? I don’t know. Nor can we see the level of change in the start dates of the breeding period, spawn, land uses or the values of the variables used. Basic data are missed.
Secondly, the presentation of the results is somewhat cryptic and difficult to understand. There is no factor score tables from any of the principal component analyses. Perhaps correspondence analysis could be more visual. Authors don’t include any significance levels of the estimated correlations or models given.
Thirdly, there is a great deficiency in the bibliography consulted. There is a lot of work that needs to be included in the introduction and especially in the discussion.
Most studies use day of the year (DOY) as a variable to test how the start of breeding success varies in different years and to analyses its association with climatic variables. Only in Suppl. Mat. Use this variable but it isn’t explained I methodology.
I believe that in its present form the manuscript cannot be accepted. It needs a very deep restructuring and a greater and clearer development of the analysis of the variables used. The complete lack of tables with raw data makes it impossible to verify how the analyses have been carried out. It is also not detailed how the models were developed and when the predictor variables were used.

See specific comments in the pdf

Reviewer 2 ·

Basic reporting

Overall the article is well-presented. The introduction presents concise background to the work, but I feel the rationale and primary context of the work could be better evidenced. For example, there are several (UK-based) R. temporaria studies which are directly relevant to the research question which have not been cited in the introduction. I list these in the attached general comments PDF. There is there is also some ambiguity the Data analysis subsection of the Methods with reference to how climate data were averaged.

Experimental design

The study uses data from a 21-year species monitoring programme to quantify the relative importance of chemical pollutants, land cover change and climate change on the phenology and reproductive success of the Common frog (R. temporaria) in the UK. Untangling these effects is certainly an important component of the species’ conservation and for predicting consequences of future environmental change and overall I feel the authors have done well to explore these interactions. The introduction ends with a series of predictions and/or hypotheses, but these could be more clearly distinguished as such. The methods mostly include all the necessary information, but I’ve suggested some re-organisation of the content. Unfortunately, the code used to carry out the analyses was not available via the link provided so I cannot comment on this.

Validity of the findings

The data on which the conclusions are based are that of a systematic long-term monitoring programme with high spatial and temporal precision. The main weakness is that of a relatively low number of sites and some sites with lacking annual replication which could be acknowledged in the Discussion section of the report. The conclusions link to the original research question as the authors report clear associations between the timing of R. temporaria breeding activity and a gradient of climatic conditions. However, from my understanding, the current analysis does not reveal “shifts” or “advancements” in phenology as the authors suggest. An explicit analysis which quantifies year-to-year variation in the timing of breeding events, which demonstrate a time-series of advancing annual phenology, would be required to make these claims. This discrepancy may simply be due to erroneous use of terminology. However, if the authors did perform this analysis, I feel they should make this clearer in the reporting of their Data analysis and Results sections.

Additional comments

Please see attached PDF for general comments.

Annotated reviews are not available for download in order to protect the identity of reviewers who chose to remain anonymous.

---

## Round 0.2 · accepted · Accept

Thanks for including more citations / links to the literature, and editing for clarity -- the reviewer and I think this paper is much improved.

·

Basic reporting

Oliver and Harrison PeerJ.
The authors have carried out a deeper review of the article and have adjusted to the reviewer's recommendations. In its current form the study can be published, with only a small change to be made. Most references do not include the DOI. This deficiency must therefore be corrected. For the rest, the work can be published in its current form pending what the other reviewer and the editor decide.

Experimental design

It has been modified according to the reviewer's recommendations

Validity of the findings

In its current form, the data and analyzes support the results obtained and the conclusions reached.

Additional comments

Ony add the DOI in the references.